# Learning from Fragmentary Multivariate Time Series Data with Scalable Numerical Embedding

## Abstract

The recent proliferation of transformer-based models in natural language processing and computer vision has significantly impacted fields involving multivariate time series (MTS) data. This research focuses on a different data type sourced from electronic health records (EHR). Unlike other MTS data, EHR exhibits a high prevalence of irregular missing values due to its asynchronous measurement nature, which may drastically harm the efficacy of the learning algorithms. To tackle this issue effectively, we propose a novel approach termed *SCAlable Numerical Embedding* (SCANE), which treats each value as an independent token to enhance the flexibility of the interaction between variables. Moreover, we integrate the transformer encoder with SCANE (TranSCANE) to form a complete feature extractor for downstream tasks. TranSCANE's attention module within its transformer encoder is specifically tailored for EHR data to circumvent the noise from irregular missing values adeptly. To further enhance the interpretability of TranSCANE, we propose the revised rollout attention that comprehensively computes attention weights across all transformer encoder stacks and neglects the dummy attention for missing values. This empowers us to gain insights into the inner workings of TranSCANE and improve model interpretability. The experimental results reinforce TranSCANE's efficacy, as it attains superior performance on three distinct EHR datasets with high missing rates. We believe that TranSCANE also holds the potential to extend the utility of transformer-based models into diverse domains with high missing rate MTS data.

## 1 Introduction

Multivariate time series (MTS) data constitutes a summary of observations registered at discrete timestamps, encompassing a multitude of interconnected variables. The pervasive influence of MTS data traverses diverse domains, including but not limited to energy, environmental, and healthcare sectors. The deployment of deep learning models for the purpose of MTS data manipulation has captured noteworthy attention in recent years (Sagheer & Kotb, 2019; Che et al., 2018; Liang et al., 2022; Zerveas et al., 2021). However, previous works predominantly employed timestamp-level approaches to process MTS data, where variables at identical timestamps are utilized as model inputs. While timestamp-level modeling inherently emphasizes temporal relations between different timestamps, it may inadvertently neglect the spatial (feature-wise) relationships among variables.

Furthermore, dealing with missing values has posed a persistent challenge in utilizing MTS data within specialized domains. One such example is the realm of electronic health records (EHR), which sets itself apart from other MTS data sources due to its irregularly sampled and asynchronous characteristics. The irregular sampling introduces fluctuating time intervals between successive timestamps, and not all variables are captured at each timestamp. This dynamic gives rise to a substantial volume of missing values within the MTS data of EHRs. In this case, imputation is a typical technique to address missing values. For instance, GRU-D (Che et al., 2018) uses imputation with a decay mechanism and introduces a GRU-based model for handling missing values. However, in the healthcare domain, concerns arise regarding the rationale behind imputing values and the potential impact of different imputed values on downstream task models.

To address the aforementioned concerns, we propose the utilization of *scalable numerical embedding* (SCANE) based on the concept of "value as a token." This concept treats each individual value from the MTS data as a token, akin to natural language processing (NLP) works that treat each word as a token, albeit our aim is to embed continuous numerical values. SCANE performs separate embeddings for each variable at each timestamp, enabling the downstream task model to learn the temporal relation and the underlying spatial (feature-wise) relationships between variables. Also, it assigns zero vectors to missings as the embedding to avoid carrying any redundant information from the imputation to the following module. We integrate SCANE with a transformer encoder (Vaswani et al., 2017) and propose the Transformer Encoder with SCANE (TranSCANE) for several tasks, which will be detailed in later sections.

Furthermore, in the context of the healthcare domain, model interpretability assumes paramount importance. An interpretable model fosters user reassurance, enabling users to comprehend model predictions and ascertain the factors receiving the greatest attention. TranSCANE inherently possesses interpretability due to its attention mechanism, which generates an attention map revealing the attention weights between all variables. This map can interpret the interaction among all variables across spatial (feature-wise) and temporal relations. Moreover, we propose revised rollout attention, building upon the rollout attention (Abnar & Zuidema, 2020), to quantify attention weights across the entire transformer encoder stacks, specifically tailored for TranSCANE. This facilitates understanding each initial embedding's contribution from SCANE to the transformer encoder output.

In summary, the contributions of this paper can be enumerated as follows:

- Introduction of SCANE, a novel concept to address multivariate time series data.
- Application of SCANE in conjunction with the transformer encoder, showcasing its ability to circumvent the need for imputation in the presence of missing values.
- Demonstration of how deep learning models with SCANE can capture both temporal and spatial (feature-wise) relation between variables, different from the conventional timestamp-level models for MTS data.
- Proposal and interpretation of revised rollout attention specifically tailored for TranSCANE to compute attention weights across the entire transformer encoder stacks.

## 2 RELATED WORK

### 2.1 DEEP LEARNING MODELS FOR DATA FEATURING MISSING VALUES

Imputation represents a straightforward approach for handling data containing missing values, involving estimating missing entries through statistical or learning-based methodologies. Within the purview of learning-based techniques, GRU-D (Che et al., 2018) has introduced an imputation method, incorporating a learnable decay mechanism and a revised GRU model, thereby achieving state-of-the-art performance on classification tasks concerning healthcare domain data. CDSA (Ma et al., 2019) has proposed a cross-dimensional self-attention model capable of computing attention across all dimensions, encompassing time, locations, and measurements, for imputation purposes in multivariate geo-tagged time series data. IGRM (Zhong et al., 2023) has presented an interactive graph generation and reconstruction framework for tabular data imputation.

However, imputation may not be universally applicable in all scenarios. SeFT (Horn et al., 2020) explores the utility of differentiable set functions learned for data with missing values, considering the data as a set form to prevent the issue of missing values from its perspective. In our work, we take a different perspective and flexibly allow for missing values in the data. Yet, to circumvent any potential impact of imputed values on the model's predictions, we adopt a masking strategy during the forward process, ensuring that the missing entries do not influence the model's output.

### 2.2 TRANSFORMER-BASED MODEL FOR MULTIVARIATE TIME SERIES DATA

MTS data is prevalent in diverse facets of human society. In MTS data analysis, transformer-based models (Vaswani et al., 2017) have gained substantial traction. One prominent instantiation is the Time Series Transformer (TST), which proposes a transformer-based framework for MTS representation learning (Zerveas et al., 2021). Additionally, Wu et al. (2020) have employed a transformer

encoder-decoder architecture for forecasting influenza prevalence, highlighting the superior performance of transformer models compared to other deep learning and statistical models in forecasting tasks. While the aforementioned literature predominantly emphasizes the temporal relationships between variables, a limited body of research emphasizes relations between variables.

More recent research endeavors have explored the interplay of temporal and spatial (feature-wise) relationships in MTS data. For instance, SVP-T (Zuo et al., 2023) adopts a variable-position-based transformer with shape-level input for MTS classification tasks, while Spacetimeformer (Grigsby et al., 2023) treats each variable at different timestamps as an individual token fed to the transformer, resembling our core concept. However, our work extends beyond this notion by embedding each value from different features independently to address embeddings and integrating it with the transformer encoder (Vaswani et al., 2017) to build the TranSCANE, resulting in an imputation-free method for data featuring missing values and with more interpretable predictions. It is particularly suitable for many areas sensitive to imputation and interpretability, such as medicine.

## 3 METHODOLOGY

Here are the notations used in this section: $X$, an $m \times n$ matrix representing the input time series data, comprises $n$ variables at $m$ timestamps. $t_i$ denotes the timestamp associated with the $i$-th row. The sequence of timestamps $\{t_i\}_{i=1}^m$ is arranged in ascending order. All input time series data are gathered within an observation window of length $T$. $X_{i,j}$ denotes the entry of the $j$-th feature at the timestamp $t_i$.

### 3.1 SUMMARIZATION

Suppose $X$ is an irregularly sampled and asynchronous MTS for a sample. The time difference between two adjacent timestamps may not be the same, and there may be missing values in $X$. Moreover, the number of timestamps of each sample is various. The variety of the number of the timestamp makes it difficult to implement models such as GRU and Transformer practically.

To align the number of timestamps of each sample, we apply the *summarization* strategy on the input data. This strategy is first given a summarization time duration $p$ and gets $k$ $(= \lfloor T/p \rfloor)$ summarization intervals. The formed summarization intervals are $[t_1, \ t_1 + p), \ [t_1 + p, \ t_1 + 2p)$, ..., and $[t_1 + (k-1) p, \ t_1 + T)$. We then assign the rows of $X$ to the summarization intervals where their timestamps belong. Every summarization window uses the mean, the mode (the most frequently observed value), or the last observed value of the collected rows to represent the value of features in the interval. The mean is used to represent the numerical feature; the mode and the last observed value are used to represent the categorical feature. The mean and the mode are computed by dropping missing values. If there is no observation of a feature in the interval, we will assign Nan, the missing value notation, for this feature. This strategy also counts the number of rows in each summarization window and records it as an additional feature, "segment entry count," to the $X$.

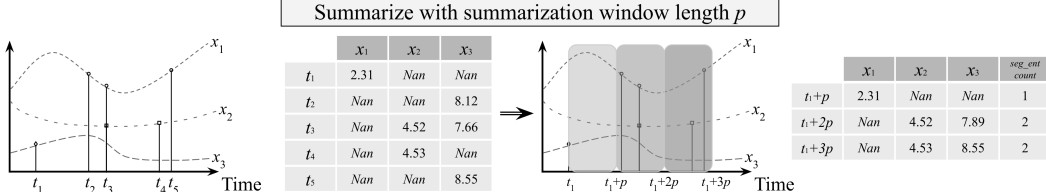

Figure 1: **Irregularly Sampled and Unsynchronized Data And Summarization**: Irregular sampling and asynchronous measurement results in a different number of variables observed per duration $p$, which is the summarization window shown in gray boxes. After performing summarization, the number and time difference of summarized observations of each sample will be identical. The "seg_ent count" is recorded after the summarization strategy, which means the number of records in the summarization window.

The input time series data will be summarized into a $k \times (n+1)$ matrix, $\boldsymbol{X}' = \begin{bmatrix} X'_{i,j} \end{bmatrix}$. The first subscript of $X'_{i,j}$ is the index of the summarization window, and the second subscript is the feature indicator. We defined a $k \times (n+1)$ missing mask matrix $\boldsymbol{M} = [M_{i,j}]$ to indicate the position of observed values in $\boldsymbol{X}'$ (with $1's$):

$$M_{i,j} = \mathbf{1}_{\{X'_{i,j} \text{ is not missing}\}}.$$

After constructing the missing mask matrix, we impute the Nan in $\boldsymbol{X}'$ with the corresponding features' global mean (for numerical features) or the global mode (for categorical features). These global mean and global mode are computed only from summarized samples in the training set. This imputation is required to implement the benchmarks for model comparison. We will show that it will not affect TranSCANE in Section 4.4.

## 3.2 Scalable Numerical Embedding

*Scalable numerical embedding* (SCANE) is a method that treats every numerical or categorical value at each timestamp in $\boldsymbol{X}'$ as a token, similar to how NLP tasks treat every word in a sentence as a token. These values are mapped to a target vector space $\mathbb{U}$ with dimension $d$. Unlike the "word to vector" approach in NLP, which only embeds tokens to the target vector space without carrying any quantity information, SCANE aims to embed both the feature concept and the feature quantity. SCANE achieves this by mapping the feature indicator to a vector in $\mathbb{U}$ and scaling the vector by the originally observed value:

$$\text{SCANE}\left(X'_{i,j}, \; M_{i,j}\right) = \left(X'_{i,j} \cdot M_{i,j}\right) f\left(j\right) = \left(X'_{i,j} \cdot M_{i,j}\right) \boldsymbol{u}_j, \tag{1}$$

where $f : \mathbb{N} \to \mathbb{U}$ is realized through a single linear layer different for each feature, and $\boldsymbol{u}_j$ is feature $j$'s feature embedding $\in \mathbb{U}$. Equation 1 shows how SCANE embeds a single variable to a vector. When SCANE gets a missing value, it will output a zero vector $\mathbf{0}^d$. To generalize SCANE to its matrix form, we have:

$$\text{SCANE}\left(\boldsymbol{X}', \; \boldsymbol{M}\right) = \begin{bmatrix} X'_{1,1}M_{1,1}\boldsymbol{u}_1 & X'_{1,2}M_{1,2}\boldsymbol{u}_2 & \dots & X'_{1,n+1}M_{1,n+1}\boldsymbol{u}_{n+1} \\ X'_{2,1}M_{2,1}\boldsymbol{u}_1 & X'_{2,2}M_{2,2}\boldsymbol{u}_2 & \dots & X'_{2,n+1}M_{2,n+1}\boldsymbol{u}_{n+1} \\ \vdots & \vdots & \ddots & \vdots \\ X'_{k,1}M_{k,1}\boldsymbol{u}_1 & X'_{k,2}M_{k,2}\boldsymbol{u}_2 & \dots & X'_{k,n+1}M_{k,n+1}\boldsymbol{u}_{n+1} \end{bmatrix}.$$

$\text{SCANE}\left(\boldsymbol{X}', \; \boldsymbol{M}\right)$ is an $k \times (n+1) \times d$ tensor. Every feature embedding in the $\text{SCANE}\left(\boldsymbol{X}', \; \boldsymbol{M}\right)$ is re-scaled by the value of the corresponding variable.

## 3.3 Positional Encoding

Our work uses the *sinusoidal encoding* as the positional encoding. We chose this fixed positional encoding because we do not want our model to learn redundant information from a trainable positional encoding module (Wang & Chen, 2020).

$$PE\left(k, i\right) = \begin{cases} \sin\left(\frac{k}{10000^{i/d}}\right), & \text{if } i \text{ is even.} \\ \cos\left(\frac{k}{10000^{(i-1)/d}}\right), & \text{if } i \text{ is odd.} \end{cases}$$

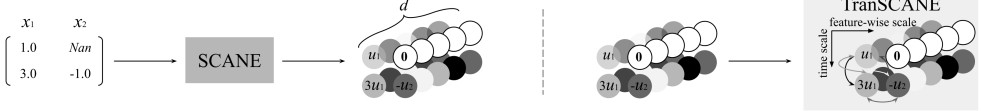

Figure 2: **Concept of SCANE and TranSCANE**: The leftmost matrix is an input time series data with $m = n = 2$. $x_1$ and $x_2$ represent two variables. Nan means missing. $\boldsymbol{u}_1$ and $\boldsymbol{u}_2$ are the feature embedding $\in \mathbb{R}^d$ of the feature $x_1$ and $x_2$, respectively. After SCANE, all observations will be mapped based on their feature embeddings proportional to the observed values, while missing values will be mapped to $\mathbf{0} = 0^d$. The right figure shows the interaction of SCANEs in the transformer encoder. TranSCANE facilitates the variable interacting flexibility across time scale and feature-wise scale.

where $k$ is index of the summarization window, $i$ is the dimension, and $d$ is the dimension of embeddings. We add these positional encodings to the SCANE. The embeddings at the identical summarization window would be added with the same positional encoding.

## 3.4 Transfomrer Encoder with Scalable Numerical Embedding

We flatten and transpose the SCANE $(\boldsymbol{X}', \boldsymbol{M})$ into a $k\,(n+1) \times d$ matrix. Similarly, we flatten matrix $\boldsymbol{M}$ into a $1 \times k\,(n+1)$ matrix. That is,

$$\bar{\boldsymbol{X}} = (\text{flatten}\,(\text{SCANE}\,(\boldsymbol{X}', \boldsymbol{M})))^T$$
$$= [X_{1,1}M_{1,1}\boldsymbol{u}_1 \quad X_{1,2}M_{1,2}\boldsymbol{u}_2 \quad \dots \quad X_{k,n+1}M_{k,n+1}\boldsymbol{u}_{n+1}]^T,$$
$$\bar{\boldsymbol{M}} = \text{flatten}\,(\boldsymbol{M}) = [M_{1,1} \quad M_{1,2} \quad \dots \quad M_{k,n+1}]\ .$$

In the transformer encoder's self-attention module, we take $\boldsymbol{Z} = \bar{\boldsymbol{X}} + PE$ to get the query $\boldsymbol{Q} = \boldsymbol{Z}\mathbf{W}_q$, the key $\boldsymbol{K} = \boldsymbol{Z}\mathbf{W}_k$, and the value $\boldsymbol{V} = \boldsymbol{Z}\mathbf{W}_v$. The $\mathbf{W}_q, \mathbf{W}_k, \mathbf{W}_v \in \mathbb{R}^{d \times d}$ are learnable parameters. To avoid paying attention to missing values, we apply the masking mechanism (Vaswani et al., 2017) to mask missings in the $\boldsymbol{Z}$.

$$\text{Attention}\,(\boldsymbol{Q}, \boldsymbol{K}, \boldsymbol{V}, \bar{\boldsymbol{M}}) = \text{softmax}\left(\lim_{\beta \to -\infty}\left(\frac{(\boldsymbol{\beta})^{k(n+1) \times 1}\,(\mathbf{1}^{1 \times k(n+1)} - \bar{\boldsymbol{M}}) + (\boldsymbol{Q}\boldsymbol{K}^T)}{\sqrt{d}}\right)\right)\boldsymbol{V}, \quad (2)$$

where $d$ is the dimension of embeddings as a suggested scaling factor (Vaswani et al., 2017) and $\beta$ is a number approaching negative infinity. $\mathbf{1}^{1 \times k(n+1)}$ is an $1 \times k\,(n+1)$ matrix whose entries are all 1. $\lim_{\beta \to -\infty}(\boldsymbol{\beta})^{k(n+1) \times 1}$ is a $k\,(n+1) \times 1$ matrix whose entries all equal to $\beta$. All attention weights with the missing as key will be suppressed by the number $\beta$, which approaches negative infinity, and they will be 0 after the $\text{softmax}$. Equation 2 shows how to use the mask to avoid paying attention to missings in the self-attention module. Moreover, due to SCANE, each value among different features and timestamps in $\bar{\boldsymbol{X}}$ can be a query and interact more with each other.

## 3.5 The Overall Architecture of the Classification Model

All deep-learning-based models in our work are composed of two modules. One is a feature extractor, and the other is a single classifier to predict the probability of classes. The classifier module contains a dense and linear layer. The dense layer contains a linear layer and GELU activation function (Hendrycks & Gimpel, 2016). We aggregate the outputs from the transformer-based feature extractor module by their means. We take the last hidden state instead of aggregation for the recurrent neural network-based feature extractor. We will compare different feature extractors in Section 4.4.

## 3.6 Revised Rollout Attention

Rollout attention is a method to simulate the information flow across all attention modules (Abnar & Zuidema, 2020). In a transformer model, the number of encoder stacks is usually multiple. It is unreasonable to take only the first layer's attention weight to interpret the whole model. Rollout attention assumes the attention flow in a transformer model is linear. To get the entire attention flow in a transformer-based model, we do matrix multiplication on attention matrices from every attention module.

Suppose $\boldsymbol{W}_i$ is the attention weight from the TranSCANE's $i$-th stacked attention module in equation 2, and there are $N$ stacks in the transformer encoder. It defines the raw attention $\boldsymbol{A}_i = 0.5\boldsymbol{W}_i + 0.5\boldsymbol{I}$ to reflect the residual connection, where $\boldsymbol{I}$ is the identity matrix. The parameter 0.5 is used to normalize the raw attention. With the raw attention from each attention module, the rollout attention $\tilde{\boldsymbol{A}}$ is:

$$\tilde{\boldsymbol{A}} = \boldsymbol{A}_N \cdot \boldsymbol{A}_{N-1} \cdot ... \cdot \boldsymbol{A}_2 \cdot \boldsymbol{A}_1\ .$$

The rollout attention $\tilde{\boldsymbol{A}}$ reflects how a transformer encoder emphasizes initial embeddings. It can be somehow considered as the feature importance for each input variable. However, the original rollout attention does not discard the missing values. It is affected by the residual connection of missing

values' zero vectors that shall have no practical impact. We revise the raw attention $\boldsymbol{A}_1$ in the $\tilde{\boldsymbol{A}}$ to fit our application with significant missing values. The revised raw attention $\boldsymbol{A}_1'$ is defined as:

$$\boldsymbol{A}_1' = \text{norm}\left(\boldsymbol{W}_1 + \text{diag}\left(\bar{\boldsymbol{M}}\right)\right), \tag{3}$$

where $\text{norm}$ is the normalization function in terms of row and $\text{diag}$ is an operator for constructing a diagonal matrix whose diagonal entries are $M_{1,1}, M_{1,2}, \ldots, M_{k,n}$, and $M_{k,n+1}$. The modified rollout attention for our work becomes:

$$\tilde{\boldsymbol{A}}' = \boldsymbol{A}_N \cdot \boldsymbol{A}_{N-1} \cdot \ldots \cdot \boldsymbol{A}_2 \cdot \boldsymbol{A}_1' .$$

We only need to revise the raw attention for the first attention module to block the weights corresponding to missing values. With this modification, the attention to missing value embeddings (zero vectors) will not be considered throughout the rollout attention calculation. The reason is as follows. First, in equation 3, the elements in $\boldsymbol{W}_1$ belonging to the missing values' queries are set to zero through operations in equation 2. Second, the residual connections belonging to the missing values are set to zero by $\text{diag}\left(\bar{\boldsymbol{M}}\right)$. Combining these two, all paths involving the impact of missing values are excluded completely from the first step of the calculation (i.e., $\boldsymbol{A}_1'$). Since the rollout attention is calculated in a cumulative and sequential sense, the terms considered in the subsequent calculations ($\boldsymbol{A}_N \cdot \boldsymbol{A}_{N-1} \cdot \ldots \cdot \boldsymbol{A}_2$) after the first step will no longer involve components from missing values. This is consistent with the fact that missings are embedded to zero vector with SCANE and would not leave any information via the residual connection of the transformer encoder module.

## 4 EXPERIMENTS AND RESULTS

### 4.1 DATASETS

For robustness, we conduct experiments on three distinct datasets: the Anonymous Hospital Hepatocellular Carcinoma Dataset (private), PhysioNet2012 (public), and MIMIC-III (public). All three datasets originate from the healthcare domain and are characterized by irregular sampling and unsynchronized measurement, thereby presenting challenges for MTS classification tasks.

#### 4.1.1 ANONYMOUS HEPATOCELLULAR CARCINOMA DATASET (HCC)

This private dataset is sourced from the Anonymous Hospital and comprises records from patients over a one-year-length observation window since patients' first diagnosis record. It is collected under Institutional Review Board (IRB) approval. The dataset includes 27 numerical features (e.g., alanine aminotransferase and alpha-fetoprotein) and 10 categorical features (e.g., Anti-HCV and HBsAg), which are listed in Appendix A. The primary objective of this dataset is to predict whether a patient will develop hepatocellular carcinoma within the ensuing five years. The dataset exhibits pronounced class imbalance with 1523 positive and 32773 negative samples, indicating an imbalance ratio of 0.046. After the summarization with a summarization window length ($p$) of 90 days, the average missing rate of all features amounts to 0.7464, a remarkably high proportion of missing values. We perform a stratified train-test-split for model evaluation to divide samples into training and testing sets with a ratio of 8:2.

#### 4.1.2 PHYSIONET2012 (P12)

The public dataset is derived from the 2012 PhysioNet challenge (Goldberger et al., 2000), encompassing 11988 intensive care unit (ICU) stays lasting at least 48 hours. The central task for this dataset is to predict if the patient dies during their hospital stay. The dataset exhibits class imbalance with 1707 positive samples and 10281 negative samples, with an imbalance ratio of 0.142. The dataset consists of 40 numerical features (e.g., glucose and urine) and 2 categorical features (e.g., sex and ICU type), detailed in Appendix A. The observation window spans 48 hours, with a summarization window length ($p$) set to 2 hours. After summarization, there are 24 summarization windows, and the average missing rate of all features is 0.7377. We adopt the train-test-split strategy from Horn et al. (2020).

#### 4.1.3 MIMIC-III (MI3)

This public dataset comprises numerous ICU patients with laboratory test results, encompassing 13 numerical features (e.g., heart rate and oxygen saturation) and 4 categorical features (e.g., Glasgow

coma scale eye-opening and Glasgow coma scale total), as enumerated in Appendix A. Preprocessing of the data follows Harutyunyan et al. (2019)'s work. The binary classification task for this dataset entails predicting patients' survival during their hospital stay. The observation window for this task spans 48 hours after patients' initial hospitalization, with the summarization window length ($p$) set to 2 hours. The dataset comprises 21107 samples, consisting of 2791 positive and 18316 negative samples, with an imbalance ratio of 0.132. We adopt the train-test-split strategy from Harutyunyan et al. (2019). After summarization, the average missing rate of all features in the summarized data is 0.4423.

## 4.2 BASELINE MODELS

We have selected a set of models to compare against our proposed model, TranSCANE. The non-sequential benchmarks encompass Random Forest (Breiman, 2001) and XGBoost (Chen & Guestrin, 2016), while the deep-learning-based benchmarks include GRU (Chung et al., 2014), GRU-D (Che et al., 2018), and the original Transformer Encoder (Vaswani et al., 2017). These models have garnered widespread use in healthcare, particularly with EHR data.

For Random Forest and XGBoost, we have employed their empirically optimal default hyperparameters. The data for these two models is subjected to a *summarization* strategy, wherein the missing mask is concatenated to the original data by timestamps. Before being fed to the model, the sequential data is flattened into a one-dimensional matrix.

All the deep learning models adhere to a common architectural structure, depicted in Section 3.5. These models comprise a feature extractor and a classifier, consisting of a dense layer followed by a linear layer, while the feature extractors vary across these models. Specifically, GRU utilizes a GRU-based feature extractor, GRU-D employs a GRU-D-based feature extractor, and the transformer encoder adopts a transformer-encoder-based feature extractor. The *summarization* strategy is also applied to these models, with the sequence data for GRU and transformer encoder being concatenated with the missing mask for each timestamp.

Our proposed model, TranSCANE, adheres to the same architectural structure outlined in Section 3.5. It employs a transformer-encoder-based feature extractor with SCANE. Since we have both numerical and categorical features, we form two separate SCANE modules for each. Notably, these SCANE modules comprise different linear transformations for obtaining individual features' embedding. Also, in the transformer encoder, we effectively mask missing values across all transformer encoders using the mask derived after *summarization*. This approach ensures that TranSCANE is robust enough to exclude missing input data values.

## 4.3 EXPERIMENTAL SETUP AND EVALUATION METRICS

To ensure a fair comparison, all deep models are trained under a uniform framework and on the same platform (detailed in Appendix C). Given the inherent class imbalance in the datasets, we adopt focal loss (Lin et al., 2018). To optimize the performance of the models, we employ a grid search strategy, with the search ranges for each model's hyperparameters provided in Appendix B. We select hyperparameters according to models' performance on the validation set, which constitutes 20% of the training set. Throughout the training process, we monitor the model's performance on the validation set every 5 epochs and halt the process if there is no improvement in the area under the precision-recall curve (AUPRC) or the area under the receiver operating characteristic curve (AUROC) for a continuous span of 30 epochs. The batch size is fixed at 256 for all experiments. The maximal training epochs for the `HCC`, `P12`, and `MI3` are set to 100, 500, and 400, respectively, ensuring adequate training for each dataset to capture underlying patterns and achieve convergence.

Because all of the datasets we used are imbalanced, we take the AUPRC as the primary metric to evaluate each model's performance. AUPRC is more indicative of an imbalanced binary classification task than AUROC (Saito & Rehmsmeier, 2015). We also adopt AUROC and concordance index (c-index) as auxiliary metrics. However, the event time is not contained in `P12` and `MI3`, so the c-index is substituted with accuracy on these two datasets to evaluate models. The decision threshold for computing accuracy here is fixed at $0.5$.

| Dataset | HCC | | | P12 | | | MI3 | | |
|---|---|---|---|---|---|---|---|---|---|
| Metric | AUPRC | AUROC | c-index | AUPRC | AUROC | accuracy | AUPRC | AUROC | accuracy |
| Random Forest | 0.3934 (0.0583) | 0.8705 (0.0232) | 0.8637 (0.0227) | 0.4805 (0.0533) | 0.8270 (0.0228) | 0.8663 (0.0146) | 0.4367 (0.0517) | 0.8319 (0.0209) | 0.8965 (0.0105) |
| XGBoost | 0.3887 (0.0592) | 0.8714 (0.0215) | 0.8644 (0.0209) | 0.4980 (0.0544) | 0.8453 (0.0203) | 0.8708 (0.0140) | 0.4553 (0.0527) | 0.8247 (0.0209) | 0.8968 (0.0105) |
| GRU | 0.4209 (0.0579) | 0.8991 (0.0156) | 0.8915 (0.0152) | 0.5222 (0.0571) | 0.8573 (0.0196) | 0.8750 (0.0138) | 0.4971 (0.0502) | 0.8537 (0.0203) | **0.9012** **(0.0107)** |
| GRU-D | 0.4519 (0.0571) | **0.9012** **(0.0171)** | **0.8934** **(0.0167)** | 0.5314 (0.0575) | 0.8524 (0.0215) | **0.8804** **(0.0135)** | 0.4752 (0.0551) | 0.8415 (0.0214) | 0.8959 (0.0105) |
| Transformer Encoder | 0.4139 (0.0571) | 0.8964 (0.0171) | 0.8888 (0.0171) | 0.5435 (0.0560) | 0.8572 (0.0200) | 0.8767 (0.0131) | 0.5074 (0.0510) | **0.8606** **(0.0187)** | 0.8953 (0.0105) |
| TranSCANE | **0.4553** **(0.0577)** | 0.8943 (0.0179) | 0.8867 (0.0179) | **0.5504** **(0.0563)** | **0.8602** **(0.0197)** | 0.8783 (0.0129) | **0.5233** **(0.0511)** | 0.8492 (0.0205) | 0.8910 (0.0104) |

Table 1: This shows the overall results of each model on the three test sets.

**Predicted Probability with Different Imputation**

| AFP Imputed Value | 1 | 10 | 100 |
|---|---|---|---|
| Random Forest | 0.1400 | 0.1600 | 0.2400 |
| XGBoost | 0.0133 | 0.7655 | 0.9741 |
| GRU | 0.3107 | 0.3656 | 0.4573 |
| GRU-D | 0.2216 | 0.2320 | 0.3242 |
| Transformer Encoder | 0.2200 | 0.3177 | 0.4371 |
| TranSCANE | 0.1389 | 0.1389 | 0.1389 |

Table 2: This table shows the predicted probability from each model under different imputations for missing values.

**Performance on The Timestamp-shuffled Samples**

| Model | HCC | P12 | MI3 |
|---|---|---|---|
| Random Forest | 0.3240 | 0.4297 | 0.3926 |
| XGBoost | 0.3228 | 0.4215 | 0.3805 |
| GRU | 0.3803 | 0.5010 | 0.4198 |
| GRU-D | 0.4094 | 0.4132 | 0.3718 |
| Transformer Encoder | 0.4145 | 0.4854 | 0.4410 |
| TranSCANE | **0.4390** | **0.5039** | **0.4530** |

Table 3: This shows the AUPRC of each model on the timestamp-shuffled test set.

## 4.4 RESULTS AND DISCUSSION

**Overall Result** Table 1 depicts the performance of baseline models and our model, TranSCANE, on the HCC, P12, and MI3, respectively. We mark the best value in **boldface** and underline the second-best value for each metric. The value in the parentheses is the 95% confidence interval of the 1000 times bootstrap on the test set. Our model, TranSCANE, outperforms other models on all datasets regarding AUPRC, which is the primary metric. It also performs well on other auxiliary metrics, getting the best and second-best values in terms of AUROC and accuracy on the P12. The experimental results support that TranSCANE is a promising choice for irregular and asynchronous MTS data.

**Predicted Probability with Different Imputations** In this experiment, we investigate the impact of different imputations on the predictive probability of each model, using the trained models from the HCC dataset, identical to those presented in Table 1. We randomly select a negative sample from HCC. Intentionally, we mask the "alpha-fetoprotein" (AFP) variable at the second and fourth summarization times following the *summarization* strategy. We then impute these intentionally missing "AFP" values with different imputation values: 1, 10, and 100. Subsequently, we feed the sample with different imputations to each model and record the corresponding predicted probabilities in Table 2. The results indicate that only TranSCANE remains immune to noisy imputations and faithfully predicts the probability based on the observed values.

**Performance of Models on the Samples with Shuffled Timestamp** In this experiment, we evaluate all benchmarks and our model on the designated test sets. Unlike the preceding experiments (Section 4.4), the samples within these test sets are randomly shuffled with regard to their timestamps. Consequently, the temporal relationship of the variables is intentionally disrupted to observe whether TranSCANE can still exhibit better performance due to its flexibility to learn from multiple aspects of the data. Employing the trained models determined in the previous experiment, the results are outlined in Table 3. Evidently, TranSCANE outperforms other benchmarks across all datasets. We can also observe that compared to the results shown in Table 1, the transformer-based models maintain much better efficacy after the disturbance of temporal information. We suppose the phenomenon is due to the self-attention mechanism in the transformer encoders that helps the models focus on remaining useful information. In particular, within transformer-based models, TranSCANE

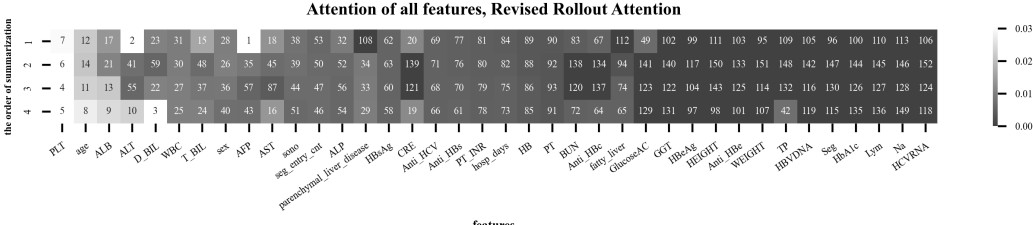

Figure 3: **Attention Weights Visualization**: In this figure, the horizontal axis represents the features, and the vertical axis stands for timestamps. The increasing brightness of a cell suggests a larger rollout attention weight and higher importance.

displays even superior performance. This outcome may allude to TranSCANE's proficiency in capturing more intricate latent interrelations among the observed variables, even when the temporal sequencing is perturbed.

**Attention Weight Visualization** We focus on a positive sample from the `HCC` for attention weight visualization. Given that we aggregate the output of the feature extractor through means, we compute the average of the $152 \times 152$ revised rollout attention (with 38 variables $\times 4$ timestamps $= 152$ features per sample under the SCANE scenario) with respect to columns. We then reshape the resulting $1 \times 152$ matrix into a $4 \times 38$ matrix, where columns and rows represent features and summarization times, respectively. Figure 3 presents the visualization of this reshaped matrix. Each cell in the figure is more illuminated if the corresponding variable receives greater attention, indicating its higher contribution to the feature extractor's output. The number on each cell denotes its attention weight's rank among all features. We rearrange the columns based on each feature's mean rank across the timestamps to facilitate readability. Notably, the top-5 features that significantly influence our model's prediction are "PLT" (platelet), "age", "ALB" (albumin), "ALT"(alanine aminotransferase), and "D_BIL" (direct bilirubin), which are highly related to hepatocellular carcinoma (Pang et al., 2015; Carr & Guerra, 2017; Yuen et al., 2009). This result strongly supports TranSCANE's interpretability, suggesting the potential for further investigation of its identified crucial factors from a medical perspective.

## 5 CONCLUSION

We introduce TranSCANE, a transformer-based model coupled with our novel SCANE embedding approach, tailored for MTS data with missing values. TranSCANE bypasses imputation for missing values and enhances its adaptability to learn latent associations among observed variables across temporal and feature dimensions. Empirical findings demonstrate that TranSCANE surpasses classic benchmarks in EHR data classification tasks. It also shows robustness against noisy imputation and temporal disruption. Furthermore, TranSCANE's interpretability is facilitated by our revised rollout attention mechanism, revealing its decision-making mechanism. In forthcoming research, we intend to extend TranSCANE's application to more domains dealing with high-missing-rate MTS data.

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

| Feature | Feature Type |
|---|---|
| AFP (Alpha-Fetoprotein) | Numerical |
| ALB (Albumin) | Numerical |
| ALP (Alkaline Phosphatase) | Numerical |
| ALT (Alanine Aminotransferase) | Numerical |
| AST (Aspartate Aminotransferase) | Numerical |
| Anti-HBc (Hepatitis B Core Antibody) | Categorical |
| Anti-HBe (Anti-Hepatitis B e-Antigen) | Categorical |
| Anti-HBs (Hepatitis B Surface Antibody) | Categorical |
| Anti-HCV (Anti-Hepatitis C Virus Antibody) | Categorical |
| BUN (Blood Urea Nitrogen) | Numerical |
| CRE (Creatinine) | Numerical |
| D-BIL (Direct Bilirubin) | Numerical |
| GGT (gamma-Glutamyltransferase) | Numerical |
| Glucose AC | Numerical |
| HB (Hemoglobin) | Numerical |
| HBVDNA (Hepatitis B Virus DNA) | Numerical |
| HBeAg (Hepatitis B e-Antigen) | Categorical |
| HBsAg (Hepatitis B Surface Antigen) | Categorical |
| HCVRNA (Hepatitis C Virus RNA) | Numerical |
| HbA1c (Glycated Haemoglobin) | Numerical |
| Lym (Lymphocyte) | Numerical |
| Na (Sodium) | Numerical |
| PLT (Platelet) | Numerical |
| PT (Prothrombin Time) | Numerical |
| PT INR (PT International Normalized Ratio) | Numerical |
| Seg (Neutrophils) | Numerical |
| T-BIL (Total Bilirubin) | Numerical |
| TP (Total Protein) | Numerical |
| WBC (White Blood Cell) | Numerical |
| Height | Numerical |
| Weight | Numerical |
| fatty_liver | Categorical |
| paranchymal_liver_disease | Categorical |
| Age | Numerical |
| hosp_days | Numerical |
| Sex | Categorical |
| sono | Categorical |

Table 4: **Feature in Anonymous Hepatocellular Carcinoma Dataset.**

APPENDIX

A    FEATURES IN THE DATASETS

Table 4 to 6 list the full feature set of the datasets applied. In Table 4, "fatty_liver" is a categorical feature to show the fatty liver severity; "parenchymal_liver_disease" is also a categorical feature to represent the severity of cirrhosis; "hosp_days" is the number of hospitalization days; "sono" represents whether a patient has the abdominal ultrasound imaging. In Table 5, "MechVent" means whether a patient uses mechanical ventilation in the ICU.

B    SEARCH RANGE OF HYPERPARAMETERS

Table 7 shows the hyperparameter grid searching range of all deep learning models. "d_model" means the dimension of embeddings. "num_head" is the number of attention heads. "ff_dim" is the feed-forward dimension of attention module in the transformer-based models. "hidden_size" is the dimension of hidden vectors in the GRU-based models. "num_layer" is the number of unit stacks.

| Feature | Feature Type |
|---|---|
| Weight | Numerical |
| ALP (Alkaline Phosphatase) | Numerical |
| ALT (Alanine Aminotransferase) | Numerical |
| AST (Aspartate Aminotransferase) | Numerical |
| ALB (Albumin) | Numerical |
| BUN (Blood Urea Nitrogen) | Numerical |
| Bilirubin | Numerical |
| Cholesterol | Numerical |
| Creatinine | Numerical |
| DiasABP (Diastolic Arterial Blood Pressure) | Numerical |
| FiO2 (Inspired Fraction of Oxygen) | Numerical |
| GCS (Glasgow Coma Scale) | Categorical |
| Glucose | Numerical |
| HCO3 (Bicarbonate) | Numerical |
| HCT (Hematocrit) | Numerical |
| HR (Heart Rate) | Numerical |
| K (Potassium) | Numerical |
| Lactate | Numerical |
| MAP (Mean Arterial Pressure) | Numerical |
| MechVent (Mechanical Ventilation) | Categorical |
| Mg (Magnesium) | Numerical |
| PaCO2 (Partial Pressure of Carbon Dioxide) | Numerical |
| PaO2 (Partial Pressure of Oxygen) | Numerical |
| PLT (Platelets) | Numerical |
| RespRate (Respiratory Rate) | Numerical |
| SaO2 (Arterial Oxygen Saturation) | Numerical |
| SysABP (Systolic Arterial Blood Pressure) | Numerical |
| Temp (Temperature) | Numerical |
| TroponinI | Numerical |
| TroponinT | Numerical |
| Urine | Numerical |
| WBC (White Blood Cell) | Numerical |
| pH (Body Fluid) | Numerical |
| Age | Numerical |
| Height | Numerical |
| Gender | Categorical |
| ICU Type | Categorical |

Table 5: **Feature in PhysioNet2012 Dataset.**

| Feature | Feature Type |
|---|---|
| Weight | Numerical |
| Heart Rate | Numerical |
| Mean Blood Pressure | Numerical |
| Diastolic Blood Pressure | Numerical |
| Systolic Blood Pressure | Numerical |
| Oxygen Saturation | Numerical |
| Respiratory Rate | Numerical |
| Capillary Refill Rate | Numerical |
| Glucose | Numerical |
| pH (Body Fluid) | Numerical |
| Temperature | Numerical |
| Height | Numerical |
| Fraction Inspired Oxygen | Numerical |
| Glasgow Coma Scale Eye Opening | Categorical |
| Glasgow Coma Scale Motor Response | Categorical |
| Glasgow Coma Scale Total | Categorical |
| Glasgow Coma Acale Verbal Response | Categorical |

Table 6: **Feature in MIMIC-III Dataset.**

| Hyperparameter | GRU | GRU-D | Transformer Encoder | TranSCANE |
|---|---|---|---|---|
| d_model | X | X | 112, 128, 144, 160 | 112, 128, 144, 160 |
| num_head | X | X | 1 | 1 |
| ff_dim | X | X | 64, 80, 96, ..., 240, 256 | 64, 80, 96, ..., 240, 256 |
| hidden_size | 64, 80, 96, ..., 240, 256 | 16, 18, 20, ..., 48 | X | X |
| num_layer | 1, 2, 3, ..., 15, 16 | 1, 2, 3, ..., 15, 16 | 1, 2, 4, 8, 16 | 1, 2, 4, 8, 16 |
| classifier_down_factor | 2 | 2 | 2 | 2 |
| learning rate | 3e-3, 3e-4, 3e-5 | 3e-3, 3e-4, 3e-5 | 3e-3, 3e-4, 3e-5 | 3e-3, 3e-4, 3e-5 |
| Optimizer | Adam | Adam | Adam | Adam |

Table 7: **Grid Searching Range of All Deep Learning Models**

| Hyperparameter | GRU | GRU-D | Transformer Encoder | TranSCANE |
|---|---|---|---|---|
| d_model | X | X | 144 | 144 |
| ff_dim | X | X | 144 | 144 |
| hidden_size | 64 | 38 | X | X |
| num_layer | 6 | 1 | 16 | 8 |
| learning rate | 3e-4 | 3e-4 | 3e-5 | 3.00009e-5 |
| early stopping epoch | 30 | 75 | 85 | 100 |

Table 8: **The Setting of Hyperparameter in `HCC` Dataset**

| Hyperparameter | GRU | GRU-D | Transformer Encoder | TranSCANE |
|---|---|---|---|---|
| d_model | X | X | 144 | 144 |
| ff_dim | X | X | 144 | 144 |
| hidden_size | 128 | 42 | X | X |
| num_layer | 6 | 1 | 8 | 1 |
| learning rate | 3e-5 | 3e-5 | 3e-5 | 3e-5 |
| early stopping epoch | 100 | 480 | 75 | 350 |

Table 9: **The Setting of Hyperparameter in `P12` Dataset**

| Hyperparameter | GRU | GRU-D | Transformer Encoder | TranSCANE |
|---|---|---|---|---|
| d_model | X | X | 128 | 144 |
| ff_dim | X | X | 144 | 80 |
| hidden_size | 128 | 18 | X | X |
| num_layer | 6 | 1 | 8 | 1 |
| learning rate | 3e-5 | 3e-5 | 3e-5 | 3e-5 |
| early stopping epoch | 280 | 400 | 125 | 380 |

Table 10: **The Setting of Hyperparameter in `MI3` Dataset**

We use Adam optimizer Kingma & Ba (2017) to optimize all models. After the grid searching, we will do a little perturbation on the learning rate to see if the model performs better. Restricted by the GPU memory, the "num_layer" of TranSCANE on the PhysioNet2012 dataset is set to 1. We also list all settings of all models in Table 8, 9, and 10.

## C PLATFORM INFORMATION

This is the information on the platform we used to conduct all the experiments in the paper:

- CPU: Intel(R) Core(TM) i7-10700 CPU @ 2.90GHz
- Memory: 64GB
- GPU: RTX 3060 with 12GB VRAM
- CUDA version: 11.4
- gcc version: 7.5.0
- pytorch version: 1.13.1
- sklearn version: 1.1.2
- xgboost version: 1.7.5

**Predicted Probability with Different Imputation**

| ALP Imputed Value | 1 | 100 | 10000 |
|---|---|---|---|
| Random Forest | 0.4100 | 0.4500 | 0.4800 |
| XGBoost | 0.0052 | 0.0071 | 0.0106 |
| GRU | 0.3244 | 0.5086 | 0.6915 |
| GRU-D | 0.3829 | 0.4201 | 0.5266 |
| Transformer Encoder | 0.3468 | 0.3522 | 0.7291 |
| TranSCANE | 0.3550 | 0.3550 | 0.3550 |

Table 11: This table shows the predicted probability from each model under different imputations for missing values.

## D    ROBUSTNESS TO NOISY IMPUTATION

Table 11 shows a supplementary experiment for section4.4. We further randomly select a negative sample from P12 instead, whose "ALP" record is available at the first, the 20th, and the 22nd summarization intervals. We intentionally impute these missing "ALP" with 1, 100, and 10000 to observe its impact on models' outputs. Table 11 shows the robustness of TranSCANE to different (noisy) imputations. We did not perform this test on HCC since it is a private dataset and sensitive to revealing a particular sample's information.

