# OpenReview forum: "Learning from Fragmentary Multivariate Time Series Data with Scalable Numerical Embedding"
_ICLR.cc/2024/Conference — ICLR 2024 Conference Withdrawn Submission_

### Official Review · Reviewer_St5P · 2023-10-31

**Soundness:** 2 fair
**Presentation:** 2 fair
**Contribution:** 1 poor
**Rating:** 3
**Confidence:** 4

**Summary:**

This paper introduces a transformer-based model for irregular multivariate time-series data with missingness. They use scalable numerical embedding to deal with the missingness with a revised calculation of attention considering the missingness. They also modify the rollout attention method so that it is still able to provide interpretation given missingness.

**Strengths:**

They come up with revised rollout attention that can exclude the impact of missing values, which sounds interesting in terms of interpretation.

**Weaknesses:**

1. Lack of baselines in the experiment. The baselines the authors compare their proposed method are very limited. A lot of transformer-based methods have been proposed recently since the original transformer was proposed in the literature of multivariate irregular time-series, specifically in the field of model EHR data. Yet the only transformer-based model that the authors compare to is the original transformer.  Many of those methods are able to deal with the missingness. For example, Zhang et al. 2023 and many more.
2. I don’t think what the authors call “Summarization” in Section 3.1 is new. In fact, what they described is actually what a lot of people call “missingness imputation” and is already commonly used in the literature. For example, Section 3.2.1 of the same paper I mentioned earlier describes something very similar things - discretize the time into different intervals and fill each interval with some values.
3. Looking at Table 1, their proposed method does not seem to be strong enough compared to the other baseline models.

Reference:\
[1] Zhang, X., Li, S., Chen, Z., Yan, X., & Petzold, L. R. (2023, July). Improving medical predictions by irregular multimodal electronic health records modeling. In International Conference on Machine Learning (pp. 41300-41313). PMLR.

**Questions:**

I have included some of the questions in the part of Weakness. Other than that:
1. How to choose the length of the interval p? Does the choice of p have any impact on prediction? For example, a larger p would lead to a shorter summarization and a lower missing rate.
2. In Section 3.4, when flattening and transposing the SCANE tensor with the shape of $k \times (n+1) \times d$, why do you choose to convert into a $k(n+1) \times d$ matrix instead of a $k \times d(n+1)$ one? The latter makes more sense to me as $k$ is from the temporal dimension, and both $d$ and $(n+1)$ are related to embedding size.
3. Why do the authors claim their proposed method is able to “capture both temporal and spatial (feature-wise) relation between variables”? Isn’t this just the advantage of any transformer-based and RNN-based methods? Basically, I don’t believe other MTS models only capture temporal relationships as they claimed in the related work part.
4. In Section 4.4, I don’t quite understand the takeaways from the 2nd and 3rd analyses. Why are we imputing the missing entries with different values and what is the point of doing random shuffling? People are not going to do noisy imputation and random shuffling in practice anyway.  Also, since only AUPRC is reported in Table 3, I wonder whether the results from other metrics like AUROC, c-index, and accuracy are consistent.

---

### Official Review · Reviewer_GLth · 2023-10-31

**Soundness:** 2 fair
**Presentation:** 2 fair
**Contribution:** 2 fair
**Rating:** 3
**Confidence:** 3

**Summary:**

The paper proposes a method to process the Multivariate time series (MTS) data, especially for data that contains missing features due to irregularly sampling and asynchronous characteristics.
In order for the SCANE pipeline to work, the authors first propose to use summarization techniques to aggregate features from misaligned time points. The time series data will be summarized into a matrix where NaN represents no value, and its corresponding masking $M$ will contain 0 for that position. Then, SCANE will try to create a feature embedding just like each word getting a vector in NLP tasks. After flattening, the data is fed into a transformer, where the masking mechanism is used to avoid paying attention to missing values. In the attention calculation,  the weights associated with missing values approaches 0 after the softmax. To achieve interpretability, the authors apply a revised rollout attention , where the missing values are not considered during the rollout. During the experiments, the authors showcase the effectiveness of SCANE through its prediction power, and provide an example that improves model interpretability.

**Strengths:**

S1. The paper is presented in a fairly clear manner, the proper notations are used and the authors provide extensive explanations to each step of the pipeline.

S2. In the experiments, all of the dataset are well-documented. The evaluation metrics for the time series prediction sees reasonable. The authors have also considered AUPRC, AUROC, concordance index (c-index) for different evaluation metrics. The authors have also provided detailed hyperparameter searching on transformers’ architectures and learning rates.

S3. The Attention Weights Visualization provides a good example of model interpretation for MTS data, and demonstrate the effectiveness of the model.

**Weaknesses:**

W1. The reviewer’s main concern is the novelty. The summarization strategy to align time-series data is not a novel practice and have been used heavily in resolving missing values due to asynchronization. Also, transformers with the masking mechanism and rollout attention is nothing new in the community.

W2. The summarize window length P should also be an important hyperparameter to consider. If the P is too short, the summarization step can capture unique time-dependent characteristics, but may leave out more NaN values due to irregularly sampling and asynchronous characteristics. However, if the P is very long, summerization is behaving like a “max pooling” and may lose local information for that particular window. The authors have not provided any ablation study in how to resolve different P. Nor is P mentioned in the hyperparmeter grid search.

W3. In terms of writing, there are too many redundant contents in the main text. For example, fixed positional encoding measure should be excluded from the main text since people know the positional encoding formulas. In addition, the dataset descriptions are way too long to for people to read through without seeing the experimental results yet. Such contents should be removed from the main text and leave rooms for additional experiments.

W4. The experiments have only compared SCANE with Random Forest, XGBoost (with summerization strategies). GRU and transformers are the only deep learning models. The authors should have considered other MTS forecasting and prediction models, some with GNNs and some with transformers. For example, check [1], [2] and a survey paper [3].

[1]Zhang, Yunhao, and Junchi Yan. "Crossformer: Transformer Utilizing Cross-Dimension Dependency for Multivariate Time Series Forecasting." In The Eleventh International Conference on Learning Representations, 2023. https://openreview.net/forum?id=vSVLM2j9eie.

[2]Wu, Zonghan, Shirui Pan, Guodong Long, Jing Jiang, Xiaojun Chang, and Chengqi Zhang. "Connecting the Dots: Multivariate Time Series Forecasting with Graph Neural Networks." CoRR abs/2005.11650 (2020). https://arxiv.org/abs/2005.11650.

[3]Wen, Qingsong, Tian Zhou, Chaoli Zhang, Weiqi Chen, Ziqing Ma, Junchi Yan, and Liang Sun. "Transformers in Time Series: A Survey." Preprint, submitted February 2023. arXiv:2202.07125 [cs.LG]. https://arxiv.org/abs/2202.07125.

**Questions:**

Q1. In the attention weight visualization, the authors have only provided visualization with one positive sample. The reviewer is wondering whether the positive sample is cherry picked? If not, could the authors provide more samples (more of ones with missing values) to demonstrate the interpretability feature?

Q2. The reviewer is wondering how the authors select the P as the summarize window length. Have the authors considered building multiple levels of summarizations, and incorporate multiple levels to better capture the time-dependent characteristics?

Q3.  Have the authors considered to use a MTS dataset that do not have any missing features, and then generate different percentages of missing values within the dataset to demonstrate SCANE’s effectiveness in low missing data ratio and high missing data ratio scenarios?

---

### Official Review · Reviewer_9DJ5 · 2023-10-31

**Soundness:** 4 excellent
**Presentation:** 3 good
**Contribution:** 2 fair
**Rating:** 5
**Confidence:** 5

**Summary:**

This paper proposes a scalable numerical embedding (SCANE) method, which has three key contributions as featured in the paper: (i) it can handle missing values by masked attention; (ii) it allows spatial-temporal interactions along both the feature and time dimension; (iii) it utilizes the revised rollout attention to add interpretability of the model. The proposed SCANE is evaluated on one private patient hospital visit dataset and two public ICU visit datasets.

**Strengths:**

1. This paper is very practical and the technical steps are explained clearly. The data processing techniques could be useful for processing similar data.
2. The experimental evaluation on three datasets looks comprehensive and the results also show marginal improvements over baselines.

**Weaknesses:**

1. Technical contributions are not significant and some model designs need more motivations.
    - Sec 3.1: Similar techniques have been proposed in earlier papers.
    - Sec 3.2: Why this design works better? It would be better to add more justifications.
    - Sec 3.4: A very similar techinique is proposed in this recent NeurIPS2023 paper https://openreview.net/forum?id=c2LZyTyddi (the authors in the neurips paper has flattened multi-channel medical signals into a sequence and then use transformer to learn spatial-temporal relations, which is similar to the practice of flattening feature matrix in this manuscript).
2. Complexity of the model is a big concern. In Sec 3.4, the proposed model flattens and transposes the $SCANE(X', M)$ into a $k(n+1)\times d$ matrix and then applies the transformer model. Since the original transformer model has quadratic complexity, it seems that the overall model complexity will be increased a lot after this step, from O(n^2) to O(n^2k^2), which can be the major drawback of the proposed model.
3. Many important baselines are missing for the mortality prediction prediction task on two public ICU datasets. For example,

[1] Choi, Edward, Zhen Xu, Yujia Li, Michael Dusenberry, Gerardo Flores, Emily Xue, and Andrew Dai. "Learning the graphical structure of electronic health records with graph convolutional transformer." In Proceedings of the AAAI conference on artificial intelligence, vol. 34, no. 01, pp. 606-613. 2020.

4. The "Accuracy" metric seems not suitable in this extremely imbalanced setting (if the model predicts all negative, the accuracy can be more than 95%, higher than any model in the Table~1). Maybe using balanced or weighted accuracy is a better practice.

**Questions:**

1. Could you report the model running time comparison and memory usage? The flattening procedure has significantly increase the sequence length and could incur huge memory load for transformer encoder.

2. In fact, the reviewer would like to hear about the authors' opinions on the effectiveness of missing value imputation in the EHR prediction tasks. This is something related to the problem setting. As far as the reviewer knows, EHR data have a lot of missing values if we formulate them as a temporal matrix (feature cross timeline). Sometimes the missing values can be much more than the existing values, meaning that the matrix $X$ can be a sparse matrix. In these common cases, should we still formualte EHR data as a temporal matrix? Why the proposed techniques work better in this case?

3. More baselines (from the recent papers not hand-crafted by the authors) should be employed in the evaluation.

---

### Official Review · Reviewer_USDM · 2023-11-01

**Soundness:** 2 fair
**Presentation:** 2 fair
**Contribution:** 2 fair
**Rating:** 3
**Confidence:** 5

**Summary:**

The paper presents the SCAlable Numerical Embedding (SCANE) method, designed to address the challenge of missing values in Electronic Health Record (EHR) data. This approach involves converting each observed variable within a given time step into a vector. These individual variable vectors are then collectively processed using a Transformer architecture. The effectiveness of SCANE was assessed using three distinct EHR datasets.

**Strengths:**

The addressed problem is relevant for EHR data.

**Weaknesses:**

a. The concept proposed in the paper lacks novelty. The technique of numerical embedding for EHR, particularly in the context of imputation, has already been explored in several publications featured at reputable conferences and journals  [1, 2].  Furthermore, the application of the Transformer encoder, as described in the paper, is not an original concept [3, 4]. A significant oversight in the study is the lack of consideration and citation of established papers that have previously applied this approach to similar problems [1, 2, 4]. However, the well-established papers for this problem were not considered as a baseline, nor cited. Additionally, for a more rigorous baseline, the paper should have benchmarked against more established methods, such as those based on Ordinary Differential Equations (ODEs) [5]. Overall, this paper is basically reaping the idea, presented in 3, but with the mask. However, the mask idea also is not novel [6,7] before.

b. The evaluation lacks ablation studies, particularly examining the separate impacts of the variable encoding step and Transformer aggregation on performance. Additional experiments also should be considered, i.e how approach behaves with different ration of missing values.

c. The paper does not include a complexity analysis of the model, which is particularly relevant since aggregation is performed across entire (binned) time-series. This raises questions about the model's scalability and efficiency, especially when dealing with very long time-series data. Online tasks also should be considered.

d. The authors employed focal loss in their model but did not present results using cross-entropy loss for comparison. The recent studies show that it does not help [8]; so the results should be provided for all losses, if it was helpful, otherwise it is not clear.

1. Zhang et. al. "Graph-Guided Network for Irregularly Sampled Multivariate Time Series." International Conference on Learning Representations. 2021.
2. Tipirneni et. al. "Self-supervised transformer for sparse and irregularly sampled multivariate clinical time-series." ACM Transactions on Knowledge Discovery from Data (TKDD) 16.6 (2022): 1-17.
3. Horn et. al. "Set functions for time series." International Conference on Machine Learning. PMLR, 2020.
4. Shukla et. al. "Multi-Time Attention Networks for Irregularly Sampled Time Series." International Conference on Learning Representations. 2020.
5. Rubanova et. al. Latent ordinary differential equations for irregularly-sampled time series. In Advances in Neural Information Processing Systems 32, pp. 5320–5330. Curran Associates, Inc., 2019.
6. Tomasev et al. A clinically applicable approach to continuous prediction of future acute kidney injury. Nature, 572(7767):116–119, 2019.
7. Tomasev et al. Use of deep learning to develop continuous-risk models for adverse event prediction from electronic health records. Nature Protocols, 16(6):2765–2787, 2021.
8. Yèche et al. Temporal Label Smoothing for Early Event Prediction, International Conference on Machine Learning. PMLR, 2023.

**Questions:**

1. More baselines [1,3,4,5,6,7] and ablation studies should be considered, see a, b, d in Weakness;
2. Complexity analisys and performance on the online tasks.